# Sparse components distinguish visual pathways & their alignment to neural networks

**Ammar I Marvi** [*]
MIT
amarvi@mit.edu

**Nancy G Kanwisher**
MIT
ngk@mit.edu

**Meenakshi Khosla**
UCSD
mkhosla@ucsd.edu

## Abstract

The ventral, dorsal, and lateral streams in high-level human visual cortex are implicated in distinct functional processes. Yet, deep neural networks (DNNs) trained on a single task model the entire visual system surprisingly well, hinting at common computational principles across these pathways. To explore this inconsistency, we applied a novel sparse decomposition approach to identify the dominant components of visual representations within each stream. Consistent with traditional neuroscience research, we find a clear difference in component response profiles across the three visual streams—identifying components selective for faces, places, bodies, text, and food in the ventral stream; social interactions, implied motion, and hand actions in the lateral stream; and some less interpretable components in the dorsal stream. Building on this, we introduce Sparse Component Alignment (SCA), a new method for measuring representational alignment between brains and machines that better captures the latent neural tuning of these two visual systems. Using SCA, we find that standard visual DNNs are more aligned with ventral than either dorsal or lateral representations. SCA reveals these distinctions with greater resolution than conventional population-level geometry, offering a measure of representational alignment that is sensitive to a system's underlying axes of neural tuning.

## 1 Introduction

Understanding the differences in how the ventral, dorsal, and lateral visual streams process information is crucial for building good models of human vision. Extensive evidence indicates that visual information is processed along three functionally distinct cortical pathways: the ventral stream, implicated in object recognition, the dorsal stream in visually guided action, and the lateral stream in motion and social information processing (Mishkin & Ungerleider, 1982; Pitcher & Ungerleider, 2021). In order to carry out such different functions, the ventral, dorsal, and lateral streams surely represent visual information in fundamentally distinct ways: **what are the precise computations underlying these functions**, and **how do visual representations differ across the three pathways**?

Over the last decade, deep neural networks (DNNs) trained for object classification have been shown to exhibit similar internal activations (Yamins et al., 2014), functional selectivities (Blauch et al., 2022; Dobs et al., 2022), and behavioral capabilities (Dobs et al., 2023; Rajalingham et al., 2018) to the human ventral visual pathway. The hierarchy of cascading layers with shared convolutional filters was inspired by known neural architecture (LeCun et al., 1989), and indeed standard measures of similarity provide strikingly high alignment between visual representations in DNNs and the ventral stream (Yamins & DiCarlo, 2016). However, recent studies have suggested that these same and similar DNNs also capture responses in the dorsal and lateral stream similarly well (Finzi et al., 2024; Conwell et al., 2024), see also (Margalit et al., 2024). How can the same computational model capture the diversity of function across these pathways? Here we address two specific questions: first, **what—if anything—distinguishes the representations in the ventral, dorsal, and lateral streams**? And second, **why do current measures of representational alignment between brains and DNNs often fail to reflect these differences**?

---

[*]https://aimarvi.github.io/

To answer the first question we applied a data-driven approach to identify the dominant components of the fMRI response to natural images in the three visual pathways of human high-level cortex, and in intermediate layers of DNNs. We observed qualitatively distinct response profiles in the three visual streams, with often interpretable selectivities: scenes, faces, bodies, food, and text in the ventral stream; group interactions, implied motion, hand actions, scenes, and reach-spaces in the lateral stream; and scenes and implied motion in the dorsal stream. We also collected behavioral ratings to provide a rigorous, quantitative evaluation of these interpretations. These findings recapitulate known category selectivity (Kanwisher, 2010) and dissociations between the three visual pathways.

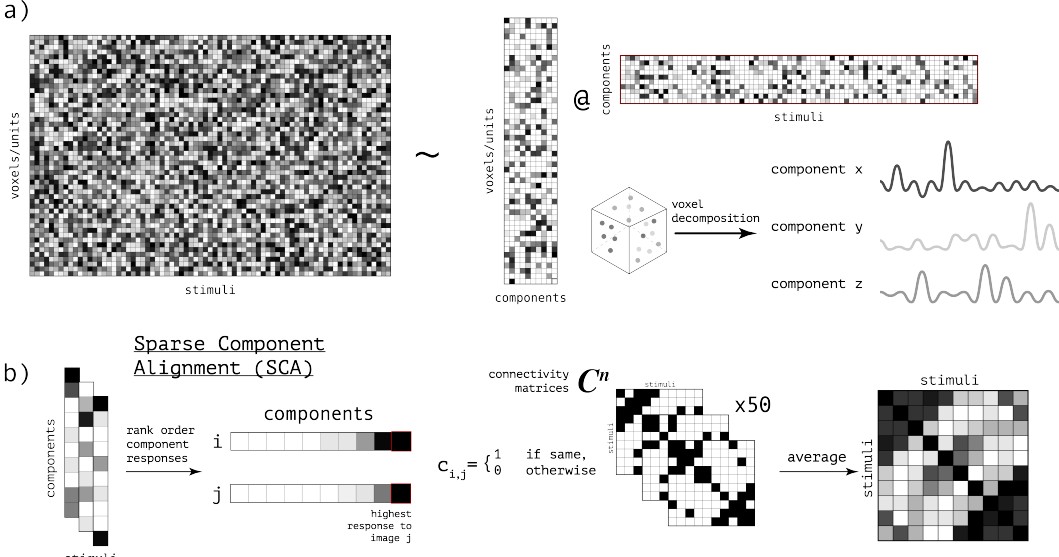

Figure 1: **Schematic overview of the data-driven component modeling approach.** (a) We used Bayesian non-negative matrix factorization (NMF) to decompose a given voxel x stimuli matrix into two lower rank matrices representing component responses $R$ and the corresponding weights of anatomical voxels $W$. (b) For each iteration, connectivity matrices $C$ are created using rank-ordered component responses, where each cell of the connectivity matrix $c_{i,j}$ represents whether a pair of stimuli $i, j$ maximally load onto the same component. Binary matrices are averaged across all iterations to produce a single image connectivity matrix.

To answer the second question, we measured brain-model alignment using two standard methods: linear encoding models (Yamins et al., 2014) and representational similarity analysis (RSA) (Kriegeskorte, 2008). Consistent with earlier work, both linear encoding and RSA suggest a high degree of alignment between the visual representations in DNNs and each of the visual pathways. However, these methods—due to inherent rotational invariances—are insensitive to the tuning properties of individual neurons, and so we developed a novel, complementary method for measuring representational alignment that retains such sensitivity. Sparse Component Alignment (SCA) uses dominant, data-driven components to capture the activity of sparse sub-populations of neurons, thus providing an alternative measure of similarity that is specific to a system's native axes of neural tuning. SCA reveals clear alignment of DNNs to the ventral visual pathway but much weaker alignment to the lateral and dorsal pathways. These findings suggest that DNNs trained on natural images better capture the computations of the ventral visual pathway, and they further raise the question of what models might better capture the lateral and dorsal streams.

## 2 METHOD

### 2.1 BAYESIAN NON-NEGATIVE MATRIX FACTORIZATION

We applied data-driven Bayesian non-negative matrix factorization (NMF) (Schmidt et al., 2009) to identify the dominant components of visual representations in each stream in four subjects of

the Natural Scenes Dataset (NSD), a massive naturalistic fMRI dataset (Allen et al., 2022). This method decomposes a neural response matrix into its dominant components, represented as the product of two lower rank matrices consisting of anatomical voxel weights for each component and the corresponding component responses. Importantly, NMF is free of any *a priori* hypotheses regarding the dataset or the spatial layout of voxels, relying solely on the data's inherent statistical structure. Our method follows a similar approach as Norman-Haignere et al. (2015) and Khosla et al. (2022) and makes minimal assumptions about the inferred components, requiring only that both matrices in the decomposition contain non-negative entries.

The neural response matrix $\boldsymbol{D} \in \mathbb{R}^{S,V}$ with $S$ images and $V$ voxels is modeled as:

$$\boldsymbol{R}\boldsymbol{W} + \boldsymbol{E} :\approx \boldsymbol{D}, \tag{1}$$

where $\boldsymbol{R} \in \mathbb{R}^{S,C}$ is the component response matrix with $C$ components, $\boldsymbol{W} \in \mathbb{R}^{C,V}$ is the voxel weight matrix, and $\boldsymbol{E} \in \mathbb{R}^{S,V}$ is a residuals matrix. Bayesian NMF introduces priors for the parameters of all these matrices ($\boldsymbol{R}, \boldsymbol{W}, \boldsymbol{E}$) to enable probabilistic inference. Specifically, following Schmidt et al. (2009), we assume a normal likelihood for the data, where the residuals $\boldsymbol{E}$ are modeled as zero-mean Gaussian noise with variance $\sigma^2$:

$$p(\boldsymbol{D}|\boldsymbol{R},\boldsymbol{W},\boldsymbol{E}) \sim \Pi_{s=1,..,S;v=1,..,V}\mathcal{N}(\boldsymbol{D}_{s,v}; (\boldsymbol{W}\boldsymbol{R})_{s,v}, \sigma^2)$$

Independent exponential priors are placed on $\boldsymbol{R}$ and $\boldsymbol{V}$, enforcing non-negativity:

$$p(\boldsymbol{R}) \sim \Pi_{s=1,..,S;c=1,..,C}\rho_{s,c}\exp(-\rho_{s,c}R_{s,c}), \quad R_{s,c} \in \mathbb{R}_{\geq 0}$$
$$\text{and} \quad p(\boldsymbol{W}) \sim \Pi_{c=1,..C;v=1,..,V}\gamma_{c,v}\exp(-\gamma_{c,v}W_{c,v}), \quad W_{c,v} \in \mathbb{R}_{\geq 0}$$

where $\rho_{s,c}$ and $\gamma_{c,v}$ are scale parameters. The posterior distributions for $\boldsymbol{R}$ and $\boldsymbol{W}$ are rectified Gaussian, while the variance $\sigma^2$ has an inverse-gamma posterior. We infer these parameters via Markov Chain Monte Carlo (MCMC) sampling.

The use of hypothesis-free methods to decompose high-dimensional data into more interpretable components has become increasingly popular with the introduction of large-scale datasets, taking advantage of statistical regularities within big data to find underlying, dominant response patterns. Common methods like principal component analysis (PCA) aim to linearly decompose data along an orthogonal set of dimensions, while others like t-SNE and autoencoders assume non-linearities to explain complex relationships within a dataset under fewer constraints.

Our motivation in using Bayesian NMF to decompose brain data is four-fold. First, unlike PCA or ICA, NMF-derived components are not constrained by orthogonality or independence, which better aligns with empirical evidence suggesting that neural responses of distinct components are often inter-dependent (Pnevmatikakis et al., 2016). Second, the non-negative constraint on matrices $\boldsymbol{W}$ and $\boldsymbol{R}$ aids in biological interpretation of fMRI data. Negative response magnitudes in $\boldsymbol{R}$ are inconsistent with neural responses in the ventral visual pathway, which usually increase after stimulus presentation, and the presence of negative values in $\boldsymbol{W}$ violates our modeling assumption that this matrix represents the relative anatomical weights of each component in every voxel. Third, while PCA is invariant to rotations in neural space, NMF is not and recovers different components before and after rotation—advantageous when modeling neural data that often consistently favor certain axes or tuning functions over others. Finally, in comparison to standard NMF algorithms, Bayesian NMF more readily discovers natural sparsity in the underlying data. Empirical studies of neuronal spiking suggest that neural responses are inherently sparse, and biological wiring costs point toward sparse connections between interacting brain regions (Olshausen & Field, 2004; Barlow, 2012; Chklovskii et al., 2002). As a result, downstream brain regions rarely have access to complete upstream neural activity. Bayesian NMF is well-suited for discovering such sparsity and—unlike standard NMF or PCA—infers sparse components that map well onto the original latent data. For these reasons—which we validate in simulated data (see Figure 2)—we opted for Bayesian NMF over other decomposition methods.

The Bayesian NMF algorithm is stochastic, so to reliably model dominant components we apply $N = 50$ iterations of the NMF algorithm for each subject. A consensus set of weight and response matrices is collected across these iterations, which is then aggregated to produce the final component weight and response matrices (see A.1 for further details). The number of components in each iteration is a free parameter, which we fix at $C = 20$ for consistency across subjects, streams, models, and layers. This was principally motivated by a previous study that used Bayesian information

criteria to estimate the optimal number of components in modeling the ventral visual stream (Khosla et al., 2022). However, we note that similar results also arise when deriving between 10 to 30 components. We then identified the most consistent components across subjects using a shared set of $1,000$ images viewed by each subject.

We similarly applied the NMF algorithm to DNN feature activations extracted in response to the same set of NSD stimuli. DNNs were pre-trained on ImageNet-1k (Deng et al., 2009) and varied in architecture and objective. Model backbones included AlexNet (Krizhevsky et al., 2012), ResNet-50 (He et al., 2016), and ViT (Dosovitskiy et al., 2020), and models were trained with category- or self-supervision (ResNet-50 w/ MoCo-v2 (Chen et al., 2020b) & SimCLR (Chen et al., 2020a), ViT w/ DINO (Caron et al., 2021)). Specifically, for each model we obtained feature activations separately for each of the four subjects' $10,000$ viewed images. The resulting unit x image matrix was treated identically to the subject voxel x image matrix for further analysis.

## 2.2 REPRESENTATIONAL ALIGNMENT

Measures of representational alignment fall into one of two broad categories: ($\mathcal{A}$) measures that establish an explicit mapping between single-neuron dimensions, and ($\mathcal{B}$) measures that compare stimulus x stimulus dissimilarities of a population (Sucholutsky et al., 2023; Harvey et al., 2023).

Linear encoding falls into the set of methods belonging to category $\mathcal{A}$ and involves linearly combining responses from model units to predict voxel responses by . This approach is well-established in the neuroscience literature, as it aims to optimally align model and brain response spaces through linear transformations while minimizing the introduction of complex non-linearities. These transformations are often preferred due to the assumption that downstream readout mechanisms apply approximately-linear functions on their inputs (Cao & Yamins, 2024). We extracted feature activations from the ultimate (AlexNet) or penultimate (ResNet-50) pooling layer of convolutional models, or from the best performing attention head in vision transformers (ViT), and used the activations to predict neural responses to a shared set of $1,000$ images viewed by each subject (via a ridge regression with an $80/20$ train/test split). Similarity between models and brains was calculated as the coefficient of determination between predicted and actual neural responses in individual subjects.

RSA belongs to category $\mathcal{B}$ and characterizes the geometry of stimulus representations in a high-dimensional neural space. We calculated the population-level dissimilarity of neural responses for each pair of the shared $1,000$ images, which we summarized in a representational dissimilarity matrix (RDM) for each subject and model. To assess the similarity between patterns of brain activity across all stimuli, we utilized correlation as our measure. We measured the second-order similarity between model representations and brain responses by extracting the upper triangular entries of each RDM and calculating the Spearman's rank correlation ($\rho$) between models and brains.

Importantly, both methods described above treat rotations as nuisance transformations and are thus explicitly insensitive to the specific tuning of individual neurons (see Figure 2). While this invariance may be useful in evaluating distributed, population-level responses, it is perhaps less desirable when representations consistently favor certain axes over others, as is the case in functional regions of interest selective for visual categories in the brain.

## 2.3 SPARSE COMPONENT ALIGNMENT

To systematically assess the consequences of rotational invariance, we propose a new technique – *Sparse Component Alignment (SCA)* – that measures representational alignment while retaining sensitivity to neural tuning. Of category $\mathcal{B}$ and similar to RSA, SCA assesses stimulus-level representational dissimilarities using Bayesian NMF-derived sparse components. However, instead of relying on population geometry, SCA computes pairwise distances between stimuli based on the likelihood that they are processed by the same dominant component.

The SCA algorithm is described in Figure 1 and Algorithm 1. For each iteration of Bayesian NMF, we decompose the neural response matrix into two lower-rank matrices representing voxel weights $\boldsymbol{W} \in \mathbb{R}^{C,V}$ and component responses $\boldsymbol{R} \in \mathbb{R}^{S,C}$. Using an overlapping set of images, we first rank-order $\boldsymbol{R}$ and then construct a stimulus x stimulus matrix $\boldsymbol{C} \in \mathbb{R}^{C,C}$, where each entry $c_{i,j}$ takes a value of one or zero based on whether that pair of stimuli produce the highest response in the same component. Mathematically, the connectivity matrices seek to partition images based on their

most contributing basis component. We define each entry of the connectivity matrix $\boldsymbol{C}$ as,

$$c_{ij} = \begin{cases} 1 & \text{if samples } i, j \text{ maximally load onto the same component,} \\ 0 & \text{otherwise} \end{cases} \tag{2}$$

These matrices are averaged across all iterations of Bayesian NMF to produce a single Image Connectivity Matrix (ICM), where the off-diagonal describes how frequently two stimuli are processed by the same dominant component.

Finally, we extract the upper triangular entries of these connectivity matrices and measure the similarity between systems using Pearson's correlation coefficient ($r$).

---

**Algorithm 1** Sparse Component Alignment (SCA)

---

1: **procedure** CONNECTIVITY MATRIX($\boldsymbol{D}$)         ▷ Neural response matrix $\boldsymbol{D}$
2:      **for** $n \leftarrow 1 : N$ **do**         ▷ # of iterations $N$
3:          $\boldsymbol{C}^n := \boldsymbol{0}^{S,S}$         ▷ # of stimuli $S$
4:          $\boldsymbol{W}_n \boldsymbol{R}_n :\approx \boldsymbol{D}$
5:          **for** $i, j \leftarrow 1 : C$ **where** $i \neq j$ **do**         ▷ # of components $C$
6:             $\boldsymbol{r}_i \leftarrow \text{rank-sort}(\boldsymbol{R}_{i,:})$
7:             $\boldsymbol{r}_j \leftarrow \text{rank-sort}(\boldsymbol{R}_{j,:})$
8:             $\boldsymbol{C}^n_{i,j} := \mathbf{1}_{\text{r}_i[0]=\text{r}_j[0]}$
9:          **end for**
10:      **end for**
11:      $\boldsymbol{C} := \frac{1}{N} \sum_{t=1}^{N} C^n$         ▷ Connectivity matrix $\boldsymbol{C}$
12: **end procedure**
13:
14: $\text{corr}(\boldsymbol{C}_A, \boldsymbol{C}_B)$

---

Leveraging the sparse decomposition of neural representations, we compare SCA with another similarity measure within category $\mathcal{A}$: the *Component Matching Score (CMS)*. This score optimizes over permutations to align components between two representations, $\boldsymbol{X}$ and $\boldsymbol{Y}$, as follows:

$$\mathcal{S}(\boldsymbol{X}, \boldsymbol{Y}) = \max_\pi (\frac{1}{C} \sum_{j=1}^{C} r_j), \tag{3}$$

$$\text{where} \quad r_j = \text{corr}(\boldsymbol{X}_{:,j}, \boldsymbol{Y}\pi_{:,j})$$

In this context, $\pi$ represents a $C \times C$ permutation matrix, allowing us to find the optimal alignment between the $C$ extracted components, accounting for possible permutations. We also measure the isolated effects of NMF pre-processing by using the components in traditional alignment metrics (see A.2 and Figure 7 for further details).

### 2.3.1 BIOLOGICAL RELEVANCE OF SCA

Many measures of representational geometry implicitly assume that downstream brain regions have access to the entire neural population response. However, biological systems face wiring constraints that limit the number of connections between regions. In reality, neurons in downstream areas can "read out" information from only a small subset of neurons in their input. By applying sparse decompositions to neural representations, we aim to distill these population-level responses into biologically interpretable components. These sparse components may offer a more realistic reflection of the information that downstream structures can access. Consequently, the image connectivity matrices—as defined above—can be interpreted as capturing the likelihood that two stimuli are to be routed to the same downstream neural structures, thus retaining sensitivity to neural tuning axes.

## 3 RESULTS

### 3.1 SIMULATIONS

Through simulations, we first demonstrate the sensitivity of our proposed SCA framework to subtle changes in the axes of representations, showing that small perturbations can significantly reduce

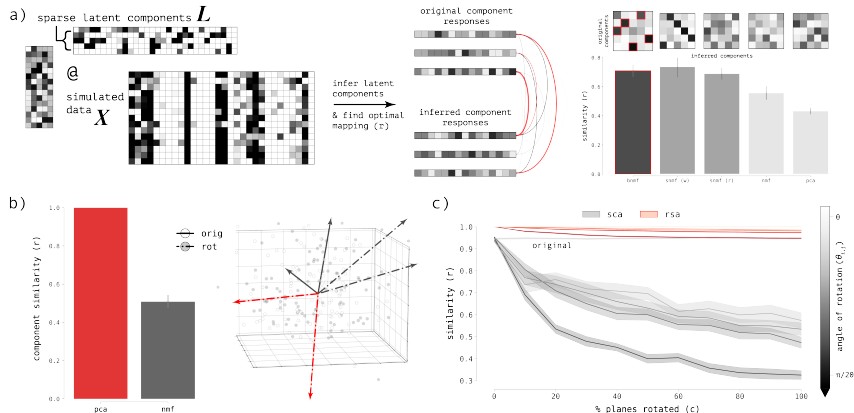

Figure 2: **Simulations of latent component recovery and rotation sensitivity** Different methods used to recover the latent components of simulated data $\mathbf{X}$. (a) A sparse decomposition finds the optimal mapping of original-to-inferred components (top, red-outlined matrix entries). Unlike sparse NMF (snmf), Bayesian NMF (bnmf) jointly infers sparsity in $\boldsymbol{W}$ and $\boldsymbol{R}$ (bottom, gray bars). (b) NMF but not PCA components are dissimilar after correcting for rotation, measured via Pearson's r. (note: overlapping PCA components) (c) Sparse component alignment (SCA) demonstrates a clear sensitivity to minor perturbations in the native axes of the representation; specifically, increasing the extent of axis rotations ($\mathbf{X} \rightarrow \mathbf{X}_r$)—whether through larger angles or a greater number of 2D planes rotated—results in more substantial decreases in alignment.

alignment. To begin, we generate simulated data as a mixture of sparse latent components and mixing coefficients. The data matrix $\mathbf{X} \in \mathbb{R}^{m,n}$ is constructed as the product of two sub-matrices: $\mathbf{L} \in \mathbb{R}^{m,k}$, which simulates the response of $k$ components to $m$ stimulus conditions, and $\mathbf{A} \in \mathbb{R}^{k,n}$, containing the mixing coefficients. The resulting equation, with added noise $\epsilon$, is defined as:

$$\boldsymbol{X} = \boldsymbol{L}\boldsymbol{A} + \epsilon \tag{4}$$

We then attempted to recover the latent components using four decomposition methods: PCA, standard NMF, sparse NMF, and Bayesian NMF (for further details, see A.3). Sparse NMF is similar to Bayesian NMF but infers sparsity in only a single sub-matrix. Figure 2 shows that a sparse decomposition indeed finds the optimal alignment between latent and inferred components.

Next, we introduce small adjustments to the native axes of representations in $\mathbf{X}$ through a rotation matrix $\mathbf{R}$ parameterized by $\theta_{i,j}$ (for details, see A.4). We then apply these rotations:

$$\mathbf{X}_r = \mathbf{X}\mathbf{R}, \tag{5}$$

and derive components using both PCA and NMF. As expected, we find that PCA but not NMF components remain unchanged after correcting for rotation (Figure 2, for details see A.5).

Finally, we study how the angle and number of 2D rotations affect the alignment between $\mathbf{X}$ and $\mathbf{X}_r$ as measured using SCA and RSA. This sensitivity analysis serves as a proof of concept, revealing that larger rotation angles ($\theta_{i,j}$) and an increased number of 2D rotations ($c$) correspond to a more pronounced reduction in alignment with SCA (Figure 2). In contrast, axis-insensitive measures like RSA deem all these rotated representations $\mathbf{X}_r$ as highly similar to $\mathbf{X}$. These findings further validate SCA's strong sensitivity to rotations in the native axes of tuning in a representation.

## 3.2 HYPOTHESIS-FREE DISTINCTION OF VISUAL STREAMS

We first applied the NMF algorithm to decompose neural responses into their dominant components using fMRI data from four subjects of the NSD. The resulting $\boldsymbol{W}$ and $\boldsymbol{R}$ matrices explained much of the variance in the neural response and performs almost as well as PCA, which sets an upper bound on the variance explained by any linear decomposition. We also empirically measured the sparsity of the resulting decomposition via the dispersion of $\boldsymbol{W}$ and $\boldsymbol{R}$ following Hoyer (2004) (Figure 3, see also A.6 and Figure 8).

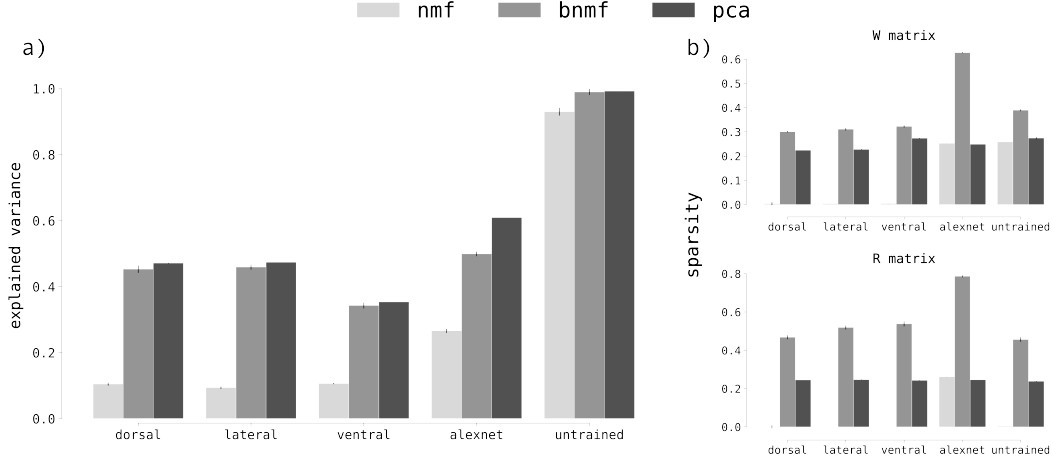

Figure 3: **Examination of data decomposition.** (a) Explained variance of an example neural response matrix $D$ in brain and models. (b) Bayesian priors produce sparse components in non-negative matrix factorization (NMF). Measured sparsity of example weight $W$ (top) response $R$ (bottom) matrices of components derivered from NMF, bayesian NMF, and PCA in the brain and models. Note: bars for standard NMF are present but close to zero.

We extracted the most consistent components (median inter-subject consistency $> 0.5$) separately in the dorsal, ventral, and lateral stream and qualitatively examined their response profiles (ie. the images producing the highest and lowest responses in each component). A majority of the response profiles offered highly interpretable category selectivities, which we further quantitatively tested with behavioral saliency rating collected online (for details, see A.7). Component selectivity in the ventral stream replicates previous studies (Khosla et al., 2022), while the components and interpretations derived for the dorsal and lateral stream that we present here are wholly novel.

The response profiles of each component is plotted in Figure 4. The same $1,000$ images are represented in each subplot via individual sticks and are colored by the average saliency rating for their respective category. Images are rank-ordered by component response magnitude (y-axis, arbitrary units (a.u.)). Alongside each sub-plot, we display three of the images producing the highest response, as well the correlation between normalized saliency ratings and component responses. We find components selective for scenes ($r = 0.632$), faces ($r = 0.799$), bodies ($r = 0.695$), food ($r = 0.604$), and text ($r = 0.439$) in the ventral stream; group interactions ($r = 0.454$), implied motion ($r = 0.660$), hand actions ($r = 0.448$), scenes ($r = 0.299$), and reachspaces ($r = 0.310$) in the lateral stream; and scenes ($r = 0.393$) and implied motion ($r = 0.428$) in the dorsal stream.

The component response profiles indicate distinct visual representations and functional roles for each of the dorsal, ventral, and lateral streams. In particular, this method refines the role of the lateral stream in social information processing, namely that separate components are selective for group interactions, hand actions, and reachspaces. We emphasize that this three-way dissociation is free of any *a priori* hypotheses regarding spatial layout and/or functional segregation, resulting only from the statistics and biases within the data and stimuli.

We also examined the response profiles of components derived from the hidden layers of various DNNs (see Figure 10). Untrained models seem selective for low-level features, while pre-trained models showed selectivities with dominant motifs for a handful of categories including faces, scenes, words, and various animals. Qualitatively, it's clear that the response profiles in DNN components are dominated by visual similarity. We note that variations in training objective and architecture did not produce notably different response profiles in the models we tested here.

### 3.3 MODEL-BRAIN ALIGNMENT

We quantified representational alignment between DNNs and each of the dorsal, ventral, and lateral streams using linear encoding, RSA, and SCA and CMS. For clarity, we focus on the alignment of pre-trained and untrained AlexNet models with the three visual streams, though the full set of models

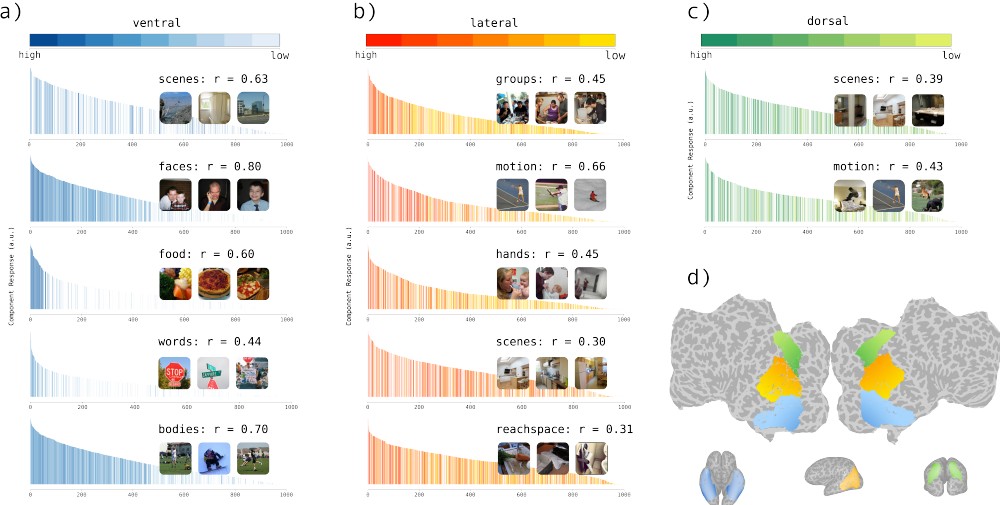

Figure 4: **Component response profiles and preferred stimuli.** Plots depicting the response profiles of the most consistent components across the four subjects in the (a) ventral (blue), (b) lateral (red), and (c) dorsal (green) streams. Each subplot shows the same $1,000$ images (depicted as sticks) rank-ordered by their evoked component response (y-axis, a.u) colored by their average saliency rating to a component-specific prompt. The correlation between saliency ratings and component responses are provided in each subplot, along with three of the component's preferred stimuli. (d) Visualization of the anatomical masks used to demarcate each visual stream.

exhibited similar patterns, as summarized in Figure 5. A linear readout suggests that pre-trained models predict neural activity similarly well in the dorsal ($r = 0.232$), lateral ($r = 0.179$), and ventral ($r = 0.180$) stream well above baseline alignment to an untrained model (dorsal: $r = 0.120$, lateral: $r = 0.096$, ventral: $r = 0.091$). When using RSA, we observed similar though notably higher levels of alignment to the ventral ($r = 0.347$) than dorsal ($r = 0.199$) or lateral ($r = 0.222$) streams. Finally, SCA suggests a markedly higher alignment between standard vision models and the ventral ($r = 0.187$) stream, and drops significantly in both the lateral ($r = 0.047$) and dorsal ($r = 0.058$) streams to levels approaching that of an untrained baseline. This measure is non-trivially related to the construction of connectivity matrices, as the same dominant components—following a strict 1-1 mapping as used in CMS—show only a modest similarity across all streams (see also A.2 and Figure 7 for further tests of NMF pre-processing).

Past studies have shown that representational alignment extends across the processing hierarchy of the ventral visual stream and DNNs. We sought to test if SCA also captured this hierarchical fit by extracting intermediate layer activations from the pre-trained Alexnet model. As expected, later layers in the model better captured neural responses in higher-level ventral visual stream, and this pattern persisted in the dorsal and lateral streams though to far lesser extent.

Altogether, these results point toward two important conclusions. First—in contrast to standard rotation-invariant measures, which suggest similar functional representations across the dorsal, ventral, and lateral streams—SCA reveals much greater alignment to just the ventral pathway. Second, this alignment is specific to models optimized for object recognition and drops for untrained models, hinting at a shared design to capture visual similarity by both DNNs and the ventral visual pathway.

## 3.4 BEHAVIORAL SIMILARITY

While alignment with neural data provides valuable insight into a system's internal mechanisms, human behavior often offers a more explicit reflection of how we represent high-level visual information. Tools like RSA are particularly well-suited to capture the similarity between neural representations and behavior, which would otherwise be difficult to quantify. We leveraged this capability to analyze the Meadows dataset—-a behavioral dataset from the NSD—-in which four participants

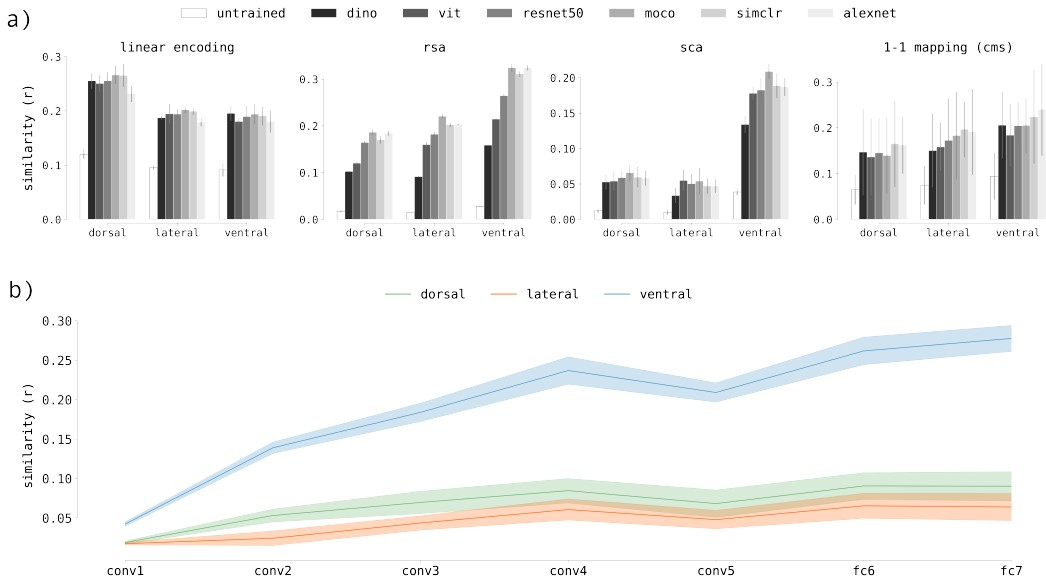

Figure 5: **Alignment of deep neural networks (DNNs) to the brain.** The measured alignment between visual representations in the brain—in dorsal, lateral, and ventral streams—and the same set of 7 visual DNNs. The untrained model is in white, and pre-trained models are colored in various shades of grey. (a) From left to right, similarity is measured by linear encoding, representational similarity analysis (RSA), sparse component alignment (SCA), and the 1-1 component matching score (CMS). (b) Alignment between each pathway and intermediate layers of a pre-trained AlexNet model, using SCA.

arranged a subset of stimuli along two dimensions based on their perceived similarity. The pairwise distances between stimuli were then used to construct a behavioral RDM.

We used either RDMs or ICMs derived from neural representations to better understand how different stimulus-level representations align with high-level behavior, quantified by computing Pearson's correlation between the corresponding similarity matrices. With RSA, behavior is most aligned with visual representations in the brain's ventral stream and in models optimized for object recognition. Alignment begins to drop for representations in the lateral stream, falls further in the dorsal stream, and is lowest in models with untrained weights. Do the connectivity matrices used in SCA capture similar patterns of information as the dissimilarity matrices used in RSA? As shown in Figure 6, analysis using connectivity matrices results in a similar overall pattern, with the ventral stream and task-optimized models showing the highest alignment to behavior. However, we don't observe the same intermediate levels of alignment with the lateral and dorsal stream. Importantly, Bayesian NMF-derived connectivity matrices seem to capture similar information as representations used in RSA while relying on a much sparser coding structure. All of this suggests that the connectivity matrices derived using Bayesian NMF effectively capture behaviorally-relevant information.

## 4 DISCUSSION

Here we sought to resolve the apparent contradiction between prior findings demonstrating (i) distinct functions of the ventral, lateral, and dorsal visual pathways in the brain, versus (ii) the similar fit of all three pathways to artificial networks optimized for object recognition. First, we used a sparse decomposition approach to identify the dominant components of visual representations in four subjects from the NSD, an fMRI dataset containing neural responses to thousands of naturalistic images. Separate analyses of the dorsal, lateral, and ventral visual pathways, as well as in a suite of DNNs trained for object recognition, revealed distinct components for each pathway. We then introduced Sparse Component Alignment (SCA) to measure the alignment of DNNs to visual representations in the brain, and we found markedly higher alignment to the ventral than either the lateral or dorsal streams. This finding is invisible to standard alignment metrics due to their rota-

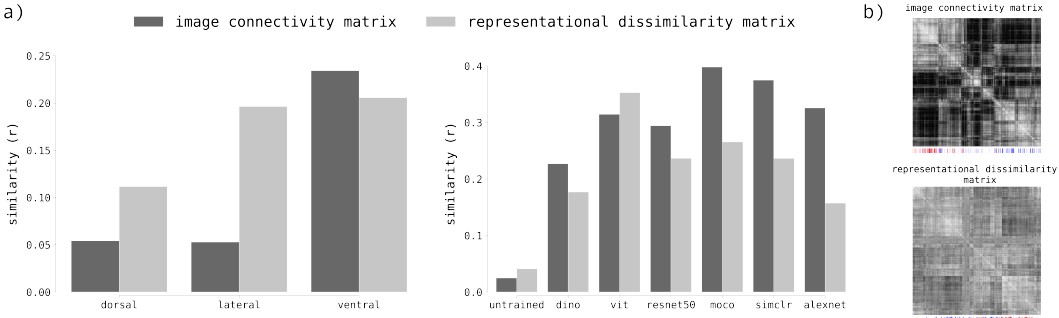

Figure 6: **Alignment to behavior.** (a) Using either representational dissimilarity matrices (light) or image connectivity matrices (dark), we measured the alignment between brains (left) and models (right) to high-level visual representations derived from the Meadows dataset. Connectivity matrices capture similar patterns of alignment while maintaining a higher degree of sparsity. (b) Both connectivity (top) and dissimilarity (bottom) matrices capture behaviorally-relevant categories. Face (red) and scene (blue) images are depicted below each matrix as colored sticks.

tional invariance. We thus conclude that DNNs share similar axes of neural tuning as neurons in the ventral visual stream.

## 4.1 FUNCTIONAL DIFFERENCES BETWEEN VENTRAL, LATERAL, AND DORSAL PATHWAYS

Using non-negative matrix factorization (NMF), we characterized neural response profiles free of *a priori* functional and/or spatial hypotheses. Consistent with prior results, we find components selective for faces, scenes, bodies, food, and words in the ventral stream. In addition, we offer interpretation of novel components selective for group interactions, scenes, hand-related actions, motion, and reachspaces in the lateral stream, and for group interactions and scenes in the dorsal stream. These results reinforce a litany of existing neuroimaging, behavioral, electrophysiological, and computational findings implicating the ventral stream in object recognition, the lateral in dynamic social perception, and the dorsal in visually guided action.

The fine-grained functional organization in the lateral and dorsal streams has remained less clear than in the ventral pathway. Our findings show that a sparse decomposition of even static snapshots is well-suited for understanding these less-explored brain regions. At the same time, the methods we used here do not fully capture the representations and computations of the dorsal and lateral streams, which would require collecting neural responses to a wider variety of stimuli and tasks.

## 4.2 ALIGNMENT OF EACH PATHWAY TO DNNS

Our finding of distinct functional roles in each pathway heightens the mystery of how all three pathways could be similarly aligned to the same image-trained DNN. We argue that standard metrics of alignment—due to rotational invariance and insensitivity to specific tuning axes—are insensitive to the functional differences we find in the neural responses across pathways. We therefore introduce SCA, a novel method that captures the tuning of neurons and network units. Using SCA, we find a pattern of results implying that units in a DNN share similar tuning properties as category-selective neurons in the ventral stream. That image-trained DNNs converge onto similar representations as the ventral pathway suggests that computations in both of these systems are driven by statistical regularities in static visual input, providing an intuitive explanation for our results. The relatively weaker alignment to dorsal and lateral representations highlights the limitations of current task objectives, architectures, and datasets, and may call for fundamentally different approaches to modeling visual pathways. Perhaps video-trained networks would better fit neural responses that are sensitive to motion, or Bayesian models of social cognition and physical scene understanding to better fit the lateral and dorsal streams, respectively. Object recognition is just one piece of the broader puzzle of human vision.

ACKNOWLEDGMENTS

This work was made possible by funding from the NIH grant EY033843 to N.G.K. and the NSF Science and Technology Center — Center for Brains, Minds, and Machines Grant CCF-1231216 to N.G.K

CODE AVAILABILITY

Code for the analyses performed in this study will be made available at `https://github.com/aimarvi/NSDstreams`.

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

# A  APPENDIX

## A.1  CONSENSUS PROCEDURE FOR AGGREGATING RESULTS ACROSS BAYESIAN NMF ITERATIONS

Following the approach of Kotliar et al. (2019) and Khosla et al. (2022), we aggregate results across $N = 50$ iterations of Bayesian NMF using a consensus algorithm. This algorithm processes the estimated component response matrices from all runs by horizontally stacking them into a large matrix, where each row represents a stimulus and each column corresponds to a component from one of the iterations. The total number of columns equals the number of iterations (50) multiplied by the number of components per iteration (20).

To ensure the stability of the components, the algorithm first identifies and removes unreliable components through an outlier detection procedure. Components are considered outliers if their Euclidean distance from the nearest neighbors exceeds a threshold of 0.8, indicating they cannot be replicated across runs.

Once outliers are removed, the remaining components from all iterations are grouped into $C$ clusters. The median of each cluster is then selected as the consensus response profile for the corresponding component.

To obtain the final voxel (or unit) weight matrix for each subject (or network), we identify—in each individual Bayesian NMF run—the component indices that show the highest correlation with the $C$ consensus component response profiles. The voxel/unit weights for these indices are normalized (to sum to 1) and averaged across runs, yielding the consensus voxel/unit weights for each component.

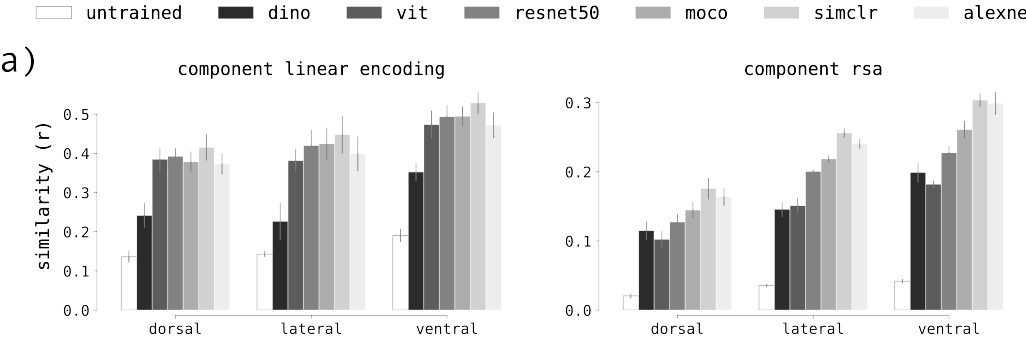

Figure 7: **Standard alignment measures using Bayesian NMF components.** Representational alignment between DNNs and the dorsal, lateral, and ventral streams, as measured by (a) linear encoding and (b) representational similarity alignment (RSA). In contrast to Figure 5, these methods use component responses in place of unit/voxel responses.

## A.2  EVALUATION OF BAYESIAN NMF PRE-PROCESSING

To further assess the effect of our chosen matrix decomposition method, we measured representational alignment using standard measures but with component—rather than unit/voxel—responses (Figure 7). Bayesian NMF pre-processing has some effect on the general pattern of alignment but does not completely account for the results given by Image Connectivity Matrices and SCA.

## A.3  SIMULATIONS ON THE RECOVERY OF SPARSE LATENT COMPONENTS

We evaluated the effectiveness of various matrix factorization techniques in recovering the true latent factors from data generated as a mixture of sparse latent components and sparse mixing coefficients. We construct the data matrix $\boldsymbol{X} \in \mathbb{R}^{m,n}$ as a product of two sparse matrices: $\boldsymbol{L} \in \mathbb{R}^{m,k}$ which simulates the response of $k$ components to $m$ stimulus conditions and $\boldsymbol{A} \in \mathbb{R}^{k,n}$, which contains the mixing coefficients. An additive noise component $\epsilon$ is included, resulting the following equation:

$$\boldsymbol{X} = \boldsymbol{L}\boldsymbol{A}^T + \epsilon$$

Here, each entry of the matrices $\{\boldsymbol{L}, \boldsymbol{A}\}$ is drawn independently from a random uniform distribution, while the noise term $\epsilon$ is sampled from a normal distribution with $\sigma = 0.01$.

Subsequently, we applied three matrix factorization methods—principal component analysis (PCA), non-negative matrix factorization (NMF), and Bayesian NMF—on the simulated data matrix $\boldsymbol{X}$, setting the number of components to $k$ for each method. To assess the similarity of the inferred component response matrix $\boldsymbol{L}'$ from each method and the ground truth latent factor matrix $\boldsymbol{L}$, we computed the following similarity score:

$$\boldsymbol{S}(\boldsymbol{L}, \boldsymbol{L}') = \max_{\pi}(\frac{1}{k}\sum_{j=1}^{k} r_j)$$
$$where \quad r_j = \text{corr}(\boldsymbol{L}_{\cdot,j}, \boldsymbol{L}'\pi_{\cdot,j})$$

In this context, $\pi$ represents a $k \times k$ permutation matrix, allowing us to find the optimal alignment between the recovered and true component matrices under permutations.

## A.4 ROTATION MATRICES

We construct the rotations in $n$ dimensions as the product of $\frac{n(n-1)}{2}$ plane rotations (Pinchon & Siohan, 2016; Clements et al., 2016; Quessard et al., 2020):

$$f(\theta_{1,2}, \theta_{1,3}, \ldots, \theta_{n-1,n}) = \prod_{i=1}^{n-1}\prod_{j=i+1}^{n} R_{i,j}(\theta_{i,j})$$

where $R_{i,j}(\theta_{i,j})$ denotes the rotation in the $i, j$ plane embedded within the $n$-dimensional representation, characterized by the angle $\theta_{i,j}$. Each 2D rotation affects only two coordinates at a time, leaving the others unchanged. For example, to rotate a 3-dimensional representation, we combine individual rotations across each 2D plane within the 3-dimensional space. By parameterizing these rotations with angles $(\theta_{1,2}, \theta_{1,3}, \theta_{2,3})$, we can express $R_{1,3}(\theta_{1,3})$ as follows:

$$R_{1,3}(\theta_{1,3}) = \begin{pmatrix} \cos\theta_{1,3} & 0 & \sin\theta_{1,3} \\ 0 & 1 & 0 \\ -\sin\theta_{1,3} & 0 & \cos\theta_{1,3} \end{pmatrix}$$

The overall rotation matrix can then be obtained by multiplying these matrices:

$$R = R_{1,2}(\theta_{1,2})R_{1,3}(\theta_{1,3})R_{2,3}(\theta_{2,3})$$

By sampling rotation matrices in this manner, we control $\theta_{i,j}$ to set the magnitude of each 2D rotation. We set $\theta_{i,j}$ to be constant across all $i, j$, choosing from $\{\pi/20, \pi/40, \pi/60, \pi/80\}$. For each of these angles, we construct the final rotation matrices by composing a varying number ($c$) of 2D rotations, where $c$ is drawn from 10 linearly spaced values between 0 and $\frac{n(n-1)}{2}$. Here, $c = \frac{n(n-1)}{2}$ corresponds to the rotation across all possible planes in an $n$-dimensional representational space.

## A.5 SIMILARITY OF ROTATED COMPONENTS

We derived components from the same set of simulated data before and after applying a set of random rotations (see A.4). We used both PCA and Bayesian NMF to derive two components from sparse, three-dimensional data, and applied the appropriate inverse transform to correct for rotation. Finally, we find the optimal mapping of original to rotated components, and measured their similarity using Pearson's r. We also measured the relationship between components using cosine similarity and found similar results.

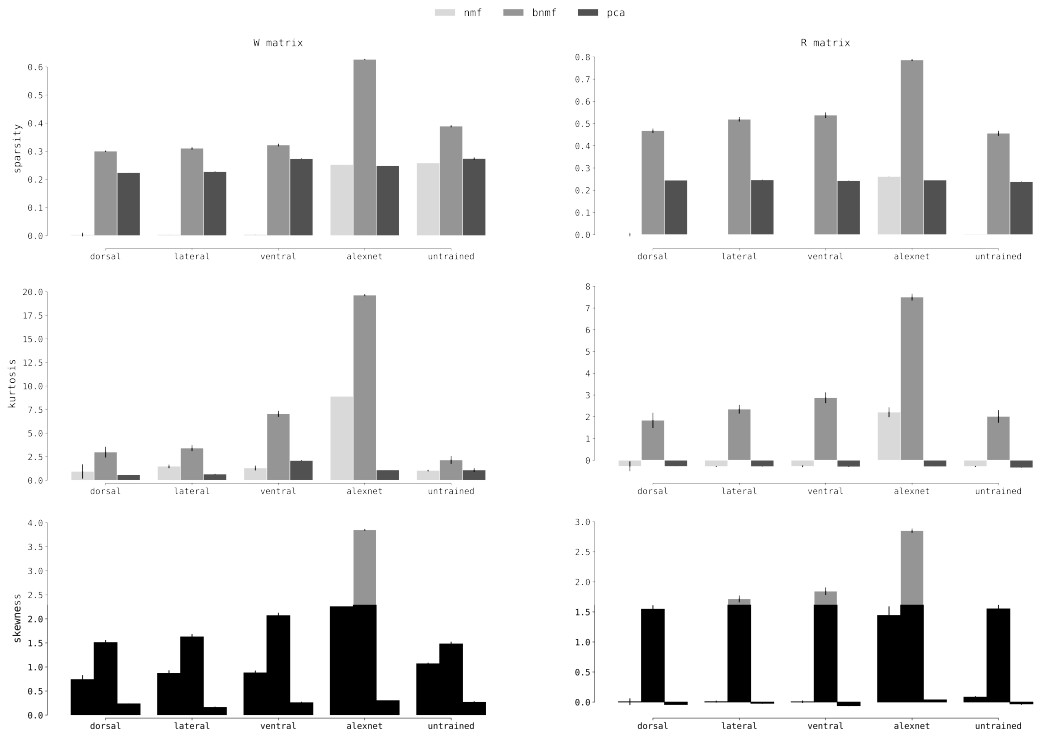

Figure 8: **Empirical measurements of sparsity in NMF decomposition.** We measured the sparsity of NMF-derived components by characterizing the distributions of $W$ and $R$ by their (a) dispersion, (b) kurtosis, and (c) skewness

### A.6 EMPIRICAL MEASUREMENT OF SPARSITY

We characterized the distribution of $W$ and $R$ for each decomposition using empirical measures of sparsity and statistical measures of kurtosis and skew.

We follow the method defined by Hoyer (2004) to measure the dispersion of a given vector $x \in \mathbb{R}^n$:

$$sparsity(\boldsymbol{x}) = \frac{\sqrt{n} - (\sum(|x_i|)/\sqrt{\sum(x_i^2)})}{\sqrt{n} - 1},$$

which measures relationships between the $L_1$ and $L_2$ norm and evaluates to 1 if $x$ contains only a single non-zero element.

We also used the $r^{th}$ sample moment $m_r$ to quantitatively measure the statistical properties of $x$. Specifically we used $m_2$, $m_3$, and $m_4$ to measure excess kurtosis $\kappa$ and skewness $\gamma$:

$$m_r = \frac{1}{n} \sum_{i=1}^{n} (x_i - \overline{\boldsymbol{x}})^r,$$

$$\kappa = \frac{m_4}{m_2^2} \quad and \quad \gamma = \frac{m_3^2}{m_2^2}$$

### A.7 BEHAVIORAL SALIENCY RATINGS

To further test our interpretations of component response profiles, we collected subjective salience ratings to the shared set of $1,000$ images viewed by all four subjects. Ratings were collected from a

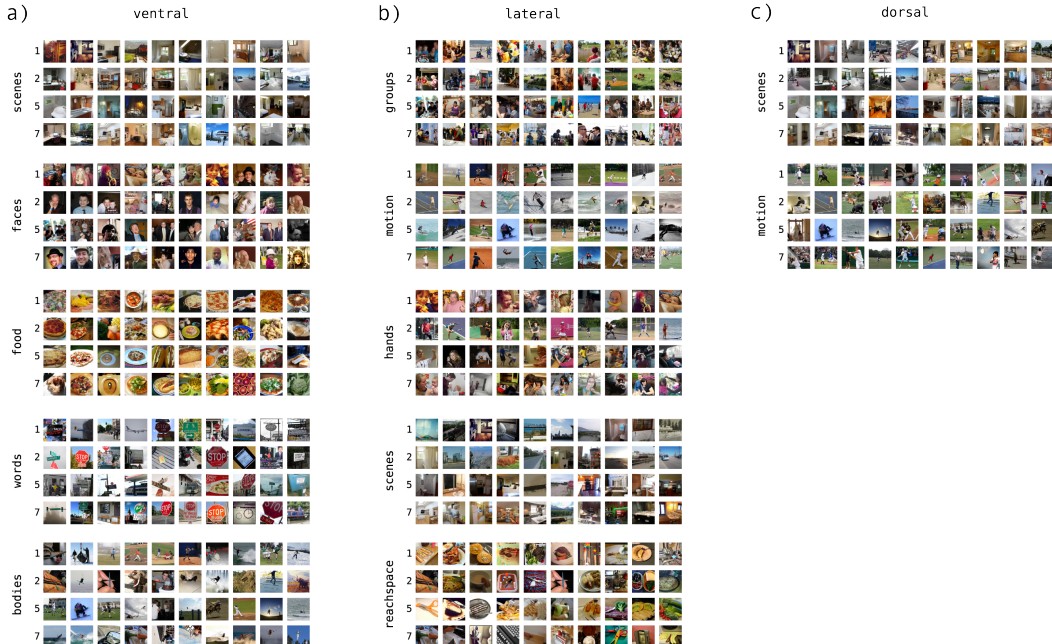

Figure 9: **Extended response profiles of neural components.** Ten of the images that produced the highest response for a given component in each of the four subjects. Response profiles are shown for the components with highest inter-subject consistency in the (a) ventral, (b) lateral, and (c) dorsal streams.

total of 125 subjects via Prolific, an online crowd-sourcing platform. In the experiment, participants were asked to rate a random sample of 250 images according to a single prompt on a Likert scale of 1 (not at all) to 7 (very much). Prior to the onset of the first trial, a prompt was randomly sampled from the following list:

1. To what extent is motion occurring in this image?
2. How prominent are hands and/or hand-directed actions in this image?
3. To what extent could you reach the contents of this image moving only your arms
4. To what extent does the image depict a place (either indoor or outdoor)?
5. How prominent in this image are groups of people interacting with each other and/or groups or people engaged in a joint activity?

The initial prompt was kept the same throughout the experiment. Images remained on-screen until the participant provided a response. Next, we displayed feedback showing the given score, followed by a 500 millisecond inter-stimulus interval.

In addition to the behavioral data we collected here, we also used a set of saliency ratings obtained in an earlier project that included prompts on scenes, faces, bodies, text, and food. Images were similarly rated by online participants, or by two independent experts in scene-selective cortex who were asked to predict how strongly the image would drive the scene-selective cortex. Further details on the experimental method can be found in Khosla et al. (2022).

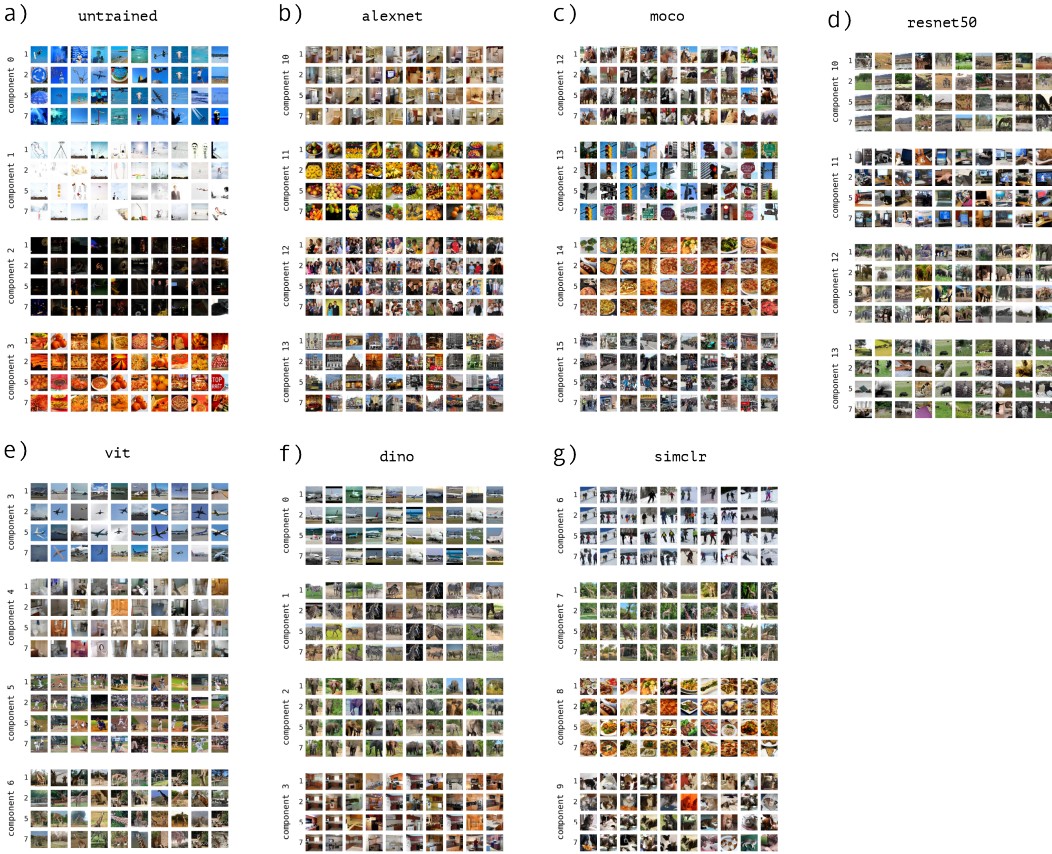

Figure 10: **Response profiles of model components.** Ten of the images that produced the highest response for a given component in each model. Separate activations were extracted for each of the four subject's $10,000$ images, leading to four sets of components for each model. Response profiles are shown for the components with highest inter-subject consistency in an (a) untrained AlexNet, and pre-trained (b) AlexNet, (c) ResNet-50 (w/ MOCOv2), (d) ResNet-50 (supervised), (e) Vision transformer (ViT, supervised), ViT (w/ DINO), and ResNet-50 (w/ SimCLR) models.

