# OpenReview forum: "Sparse components distinguish visual pathways & their alignment to neural networks"
_ICLR.cc/2025/Conference — ICLR 2025 Spotlight_

### Official Review · Reviewer_PbjK · 2024-10-26

**Soundness:** 3
**Presentation:** 2
**Contribution:** 2
**Rating:** 8
**Confidence:** 3

**Summary:**

This paper introduces Sparse Component Alignment (SCA) for quantifying alignment between two learning systems through a data-driven approach (non-negative matrix factorization). This is done by identifying dominant components that capture sparse neuronal activity. SCA is specific to a system’s native axes of neural tuning. The authors find components selective for faces, scenes, bodies, food, and words in the ventral stream. Furthermore, representations generated by different deep neural networks are found to have a greater alignment with the ventral over dorsal or lateral representations.

**Strengths:**

Quantifying model to brain similarity is an important research direction that has seen substantive progress in the last few years, and I appreciate the author’s contributions towards the same.
- The manuscript is written in a very straightforward way, and it is easy to follow.
- SCA is biologically and mathematically grounded without a priori hypotheses on neural response profiles.
- The authors show that DANNs, at least the ones tested, show a greater similarity to the ventral stream under SCA over other methods.

**Weaknesses:**

- Lines 46-9: “How can the same computational model capture the diversity of function across these pathways?” Could the authors ground this question more concretely given that the Topographic DANN (Margalit et al., 2024) was NOT trained to perform only object recognition but instead also imposed a spatial constraint on model units to find separation of visual processing into distinct streams? It was precisely topography and self-supervision that led to that outcome.
- I would have preferred seeing the TDANN’s similarity to the different streams computed using SCA, given that it was cited in the introduction. More generally, I would have preferred seeing more breadth in the computational models evaluated—what do the authors find when they use DANNs that are trained not for object recognition but instead for object detection (such as on the Pascal VOC or MS-COCO datasets) or action recognition (on the Kinetics dataset). Do these models, under SCA, show a higher (or at least somewhat better, given limitations of available neural data) match to the dorsal and lateral streams?

**Things (manuscript writing) that can be improved but did not impact my score:**
- I was having a really hard time looking at the figures since the particular font used makes visibility really low unless the figure is zoomed in a lot. It would be nice if the authors could use a darker and bolder font like Arial or Calibri for the figures.
- Line 97 (and similar phrases throughout the manuscript): “SCA reveals clear alignment of DNNs to the VVP …”. The authors should make it clearer that only those DNNs that have been (a) evaluated in this work, and (b) trained to perform object categorization. DNNs in general is a very broad term given only 4 trained networks were evaluated.

**Questions:**

How big is the ViT evaluated—is it base-16?

---

> ### Author Response · Authors · 2024-11-23
> **Response to Reviewer PbjK (1/1)**
>
> Thank you for your thoughtful feedback and thorough review of our paper. Below we respond to the questions and points of weakness you raised above:
>
>
> TDANN and self-supervision:
>
> While we do not directly evaluate the TDANN model, we analyze self-supervised models, including MOCO, which shares similar functional objectives with TDANN. To further investigate whether domain-general learning objectives like those used to train TDANN can capture responses across all three streams, we also include two other self-supervised algorithms (SimCLR and DINO). Our results show that, when using SCA for model-brain comparison, the alignment across the streams is not equal. However, linear predictivity yields similar performance across streams. This mirrors the findings of the TDANN paper, where predictivity was used to assess alignment with all three streams. The discrepancy with SCA suggests that our approach provides a more nuanced measure of model alignment with the visual streams.
>
>
> Spatial Constraints in TDANN:
>
> The major factor that distinguishes TDANN from other self-supervised learning models is the spatial constraint. However, we believe that this constraint primarily influences the model’s spatial topography rather than its functional responses, which are central to our analysis. Testing how spatial constraints might affect the nature of functional responses (beyond introducing redundancy) is an intriguing direction for future research but is outside the scope of the current study.
>
>
> Breadth of Models Evaluated:
>
> Measuring the alignment of models trained on more diverse objectives and datasets is an exciting direction for future work. In our study, we focused on object recognition models and their alignment with the three visual streams, particularly because their unexpected alignment with dorsal/lateral streams challenges the conventional view that object recognition models should primarily align with the ventral stream. However, future work should certainly explore the models you mentioned.
>
> One concern with testing models optimized for action recognition or detecting social interactions is their generally lower performance compared to object recognition models. Additionally, recent studies (e.g., Wang et al., NeurIPS 2019) suggest that models trained for other tasks, such as depth estimation or segmentation, also struggle to meaningfully distinguish between the visual streams.
>
> Other points:
>
> Figure text has been adjusted to increase readability and clarity.
> We have edited the text to more clearly state that our results extend only to the DNNs tested here.
> The ViT used here is a pre-trained base model with patch-size 16 and input resolution of (224, 224).

---

> > ### Comment · Reviewer_PbjK · 2024-11-23
> >
> > Thank you for the response. I think that addresses most of my concerns. Despite the lower performance for models trained on tasks outside of core object recognition, it would still have been informative to measure brain alignment for at least one other objective, but the manuscript is still strong regardless.

---

### Official Review · Reviewer_d41b · 2024-11-03

**Soundness:** 3
**Presentation:** 4
**Contribution:** 3
**Rating:** 8
**Confidence:** 3

**Summary:**

This paper is an attempt to solve the apparent contradiction between the observation that the ventral, dorsal, and lateral streams in human vision have different functions; and the fact that neural activity in each of these tends to be similarly correlated with deep neural-net (DNN) activity trained on a single task. The authors use Bayesian non-negative matrix factorization (NMF) to identify the different selectivities for the three visual streams from fMRI data, confirming previous results. They next introduce a different way of assessing the similarity between neural-net activations and brain recordings that finds that DNNs correlate much better with the ventral stream than with the dorsal and lateral ones.

**Strengths:**

**Originality**
* the authors observe that the rotational symmetry that is built into existing measures of representational alignment disregards the fact that biological constraints (such as the cost of neural wiring) would in fact be very sensitive to rotations that could, e.g., convert a very sparse representation (that is easy to read out with few neurons) to a very broad one.

**Quality**
* the work seems carefully done, with controls to check that the proposed methods recover known results and that they work on simulated data.

**Clarity**
* the paper is very well written with overall clear figures and detailed explanations.

**Significance**
* understanding the relation between artificial neural nets and their biological counterparts is important both for using the artificial networks to aid in understand the brain, and conversely, as a potential route to gleam inspiration from biological neural circuits to apply to artificial nets.

**Weaknesses:**

* Fig. 2 is really confusing
  * Both panels seem to mix together the question of inferring the original components correctly with the question of sensitivity to rotations.
  * I would suggest separating the two parts:
    * First, take out the "rotate" part from panel a, focus only on reconstruction, and show bar plot of similarity for different methods.
    * Second, add explanation of rotation and show its effect on SCA.
      * Might also be useful to add lines to show that NMF and PCA don't change at all when you rotate.
  * It's very confusing to have the bars colored in Fig. 2b, as the shades of gray employed there do not match the ones from Fig. 3.
    * If the shade only indicates the similarity level, then it is redundant with the height of the bars, and I would suggest getting rid of it.
*

**Questions:**

* Why is NMF worse than BNMF with simulated data (Fig. 2)?
  * ...and why is the similarity between inferred and actual components not perfect? If I understand correctly from eq. (4), the data should exactly conform to the model here.
* I don't understand the $\theta$ colorbar in Fig. 2b: what is the range actually? I see it going from $\pi/2\theta$ to $\theta$, which I can't really parse.
* Why is the NMF bar missing from most conditions in Fig. 3b?
* In section 3.2 (line 319), is it standard NMF or Bayesian that is employed? If standard, why not Bayesian?
* The color palette for the lateral stream (Fig. 4b) should go from light pink to red instead of yellow to red, to avoid a confusing change in hue (and also to be consistent with the other two panels).
* The tick and axis labels in most of the figures (Figs. 2, 3, 4, 5, 6) should be significantly bigger.

---

> ### Author Response · Authors · 2024-12-02
> **Response to Reviewer d41b (1/1)**
>
> Thank you for your thoughtful feedback and thorough review of our paper. Below we respond to the questions and points of weakness you raised above:
>
> Addressing Figure 2:
>
> The simulation in Figure 2 requires inferring the original components (L) given an observed data matrix (X). The failure to accurately reconstruct these components stem from the noisy data generating process (eq. 4), the stochasticity of NMF reconstructions, and the limitations of decomposition methods. Bayesian NMF has priors for a sparse decomposition and thus outperforms both NMF and PCA in recovering the sparse latent components in this particular simulation. For a more thorough comparison, we have also included the rotational sensitivity of RSA to Figure 2B. In addition, we are in the process of adding an analysis of rotation on NMF vs PCA components
>
> We have adjusted Figure 2 and its caption to be more clear.  The colorbar in Figure 2B measures rotation angle in radians from 0 to π/20 (ie. 0° to 9°). The confusion stems from the similarity of  ‘0’ and ‘θ,’ and so we have attempted to better distinguish the labeling of the two. The gray bars in Figure 2A now indicate which methods infer sparsity (and which method we employ in SCA).
>
> Other points:
>
> - The values for NMF in Figure 3B are present but close to zero. We have clarified this in the figure caption.
>
> - Tick and axis labels have been adjusted throughout the figures to increase legibility.

---

### Official Review · Reviewer_EmTJ · 2024-11-04

**Soundness:** 2
**Presentation:** 2
**Contribution:** 3
**Rating:** 5
**Confidence:** 3

**Summary:**

The paper introduces a new method, Sparse Component Alignment, for comparing models to brain measurements. The method is used to compare fMRI measurements from the Natural Scenes Dataset with 3 Deep neural network architectures trained on ImageNet-1k. Compared with linear (Euclidean) metrics, or RSA, the proposed method stronger alignment with the ventral stream compared with the dorsal or lateral streams.  The authors also compare to behavioral data (the Meadows dataset - sec 3.4), and find similar aligment with the ventral stream.

**Strengths:**

Development of methods for brain-model comparison is an important area. The authors have developed a promising new method, and have used it to demonstrate  that DNN recognition networks are more similar to ventral stream than dorsal or lateral. Experiments comparing the three brain regions, for 3 different DNN architectures, and 3 different similarity measures, are extensive.

**Weaknesses:**

The contributions of the paper are both methododological and scientific.  I think both have potential, but found both to be somewhat limited as presented.

- Method: The intuitive motivation for the SCA method (that it is sensitive  to changes in the axes of the representation) makes sense, but after reading it 3 times, I still don't understand how the mathematical construction of SCA leverages that, nor the extent to which the results should really be attributed to the BNMF pre-processing.  If the method is to be relied upon, I think practioners will want either a derivation tying it to some objective, or some sort of bounds on senstivity to axis changes, or an analysis of why it exhibits this sensitivity to axis changes.  Alternatively, one could provide a more thorough set of empirical tests showing the conditions under which the method works or fails.  For example, how sensitive are results to the choice of the BNMF dimensionality? The simulations provided (fig 2) are quite limited.

- Scientific result: The main result, that recognition models (across three different architectures) are more aligned with ventral stream than dorsal or lateral streams is nice, albeit expected.  RSA already provides this result, in a slightly weaker form (text on p. 8, fig 5).  It was not clear to me how to fairly compare the r-values achived by the two methods.  Nor (as mentioned above) how much the result arises from the BNMF pre-processing.  What if you compute RSA on the BNMF components?

Writing/presentation could be improved.
- please indicate up front (abstract/intro) that you're analyzing fMRI data (it was not apparent until sec 2.1).
- I think you could reduce space spent describing Bayesian NMF, which is a known method
- Provide a more thorough description of the SCA calculation - I read it several times and am still not fully understanding the construction.
- Provide a more  thorough description of the results shown in Fig 4

**Questions:**

See above.

---

> ### Author Response · Authors · 2024-11-23
> **Response to Reviewer EmTJ (1/2)**
>
> Thank you for your thoughtful feedback and thorough review of our paper. Below we respond to the points of weakness you raised above:
>
> Addressing Weakness 1 regarding the Method:
>
> The sensitivity of the SCA method to axis rotations arises from the inherent axis sensitivity of the components extracted by NMF. We first provide a mathematical explanation for why BNMF (or any NMF method) is sensitive to the axes and show how this sensitivity propagates to the SCA image × image matrix.
>
>
> NMF Axis Sensitivity:
>
> The key reason BNMF is sensitive to rotations of the feature space is the non-negativity constraint on both the components (columns of R) and coefficients (columns of w):
> Both the components and coefficients are constrained to be non-negative.
> This non-negativity constraint makes NMF non-invariant to rotations of the data axes, in contrast to methods like PCA that are rotation-invariant.
>
> When the feature space is rotated (e.g., by a linear transformation), the non-negativity constraint forces NMF to adjust the components and coefficients so that the data is still approximated by non-negative factors. This makes any NMF method particularly sensitive to the coordinate system and explains its axis dependence. We empirically demonstrate this axis sensitivity of the NMF components in Figure 2.
>
>
> Clarification of the Reviewer's Comments:
>
> We agree that it would be interesting to think about the bounds for SCA’s sensitivity to axis changes but a precise theoretical derivation may be out of scope for this manuscript. The sensitivity of NMF components (and thus, SCA) to axis changes is difficult to derive analytically, as it directly depends on how the components shift with changes in the coordinate axes, which varies with the nature of the data.
>
>
> Robustness of SCA:
>
> Additionally, we want to emphasize that the SCA method is not entirely reliant on the Bayesian NMF technique. Other NMF methods capable of recovering sparse components from the data are equally effective. We have included additional results using an alternative NMF method (Figure 2), which infers sparsity in a single sub-matrix. This method also recovers the latent components effectively, supporting our use of sparse decompositions.
>
> Lastly, we have conducted further experiments demonstrating that SCA is robust to variations in the BNMF dimensionality. When the number of components is varied (e.g., from 10 to 30), the results remain consistent, further validating the stability of the SCA method, which we will address in the revised manuscript.

---

> ### Author Response · Authors · 2024-11-23
> **Response to Reviewer EmTJ (2/2)**
>
> Addressing Weakness 2 regarding the Scientific contributions:
>
> While the higher alignment to the ventral stream is intuitive and perhaps expected, recent results seem to suggest the opposite (Finzi et al 2024, Conwell et al 2024). Even historically, recognition models have been successfully used to predict brain responses in dorsal visual stream (Eickenberg et al., 2015). Thus while neuroscientific intuitions might suggest that this result is expected, existing model-brain similarity techniques had not yet clearly differentiated the alignments of these models across the ventral versus dorsal/lateral streams.
>
> We share your question of the effect of Bayesian NMF pre-processing on the results we find here, and we attempted to address this concern with the 1-1 component matching score (Figure 5A, rightmost panel). Our conclusion from this specific analysis is that results using SCA are ‘non-trivially related to the construction of connectivity matrices’ (line 412).
>
> Nonetheless, we acknowledge that more can be done to measure the effect of NMF pre-processing on alignment to neural data. Performing RSA using component responses is an excellent suggestion and was performed earlier in our work. We find a similar pattern of results compared to traditional RSA (Figure 5A, second from left), specifically that DNNs are more aligned to the ventral stream but still weakly aligned to the dorsal and lateral streams (compared to an untrained baseline). While we originally decided not to include this analysis, we have now added it to Appendix A.2 and Figure 7). To further measure the effect of NMF pre-processing, we now also include an analysis of linear predictivity using the component responses (see Appendix A.2, Figure 7)
>
> Importantly, connectivity matrices are more biologically plausible than RDMs under the premise that connectivity matrix represents how likely 2 images are routed to the same downstream neural structure. In contrast, RDMs rely on the entire population code. However, given the anatomical and physiological constraints of the brain, downstream brain regions can likely access only a sparse subset of the representational components, rather than arbitrary components of the representational geometry. This makes connectivity matrices a more intuitive and realistic structure for analyzing image-by-image similarities in a way that aligns with how the brain processes and routes information.
>
> Connectivity matrices also are behaviorally grounded, sometimes more so than traditional RDMs, despite being much sparser than RDMs (see Figure 6 for the alignment of RDMs and connectivity matrices with behavior)
>
>
> Quantification of main results:
>
> The comparison of metrics and their proposed alignment is indeed hard to quantify. Instead we opt for a qualitative analysis and suggest that SCA can be used to complement existing methods rather than replace them. Further interpretation of such metrics can then be made by readers themselves.
>
>
> Other points:
>
> We have now updated the manuscript to make it clear that we are using fMRI data (line 053)
> Given its impact on our proposed method, we feel the time spent explaining and motivating the use of Bayesian NMF is crucial and should remain in the main text of the manuscript.
> We have attempted to clarify the description of SCA, please see the updated section 2.3.  Please note that the method is also described algorithmically and figurally, to which we have also added text to more concretely describe the steps of SCA
> We have now highlighted the results found in the lateral stream as particularly novel. However, as they rely heavily on author interpretation, we are cautious not to overstate the results from Figure 4.

---

> > ### Comment · Reviewer_EmTJ · 2024-11-29
> > **Reply to reviews**
> >
> > Thanks for your replies. My initial score was perhaps overly harsh.  And I appreciate the inclusion of an alterantive NMF method in fig 2, and the inclusion of RSA on the NMF components (appendix A2, fig 7).  Nevertheless, my concerns about the methodological motivation/justification and the scientific interpretation remain.    Some elaboration:
> >
> > Method: I appreciate the desire to regularize the comparison of model responses and fMRI data through a low-dimensional bottleneck, and I understand that NMF can provide a sparse substrate for that comparison that is axis-aligned (not rotation-invariant). The issue is with the SCA metric that is used to compare the sparse components of brain and model (sec 2.3), which seems ad-hoc. If this is to become a standard method, it will be important to provide comparison to other methods. The RSA comparison is useful (since it's a widely used method), and the NMF-component RSA is a nice addition.  But comparison to direct linear mapping (i.e., regression) is a bit of a straw man, since it is not low-D/sparse.  I would think the natural comparisons would be (1) CCA (the linear/quadratic solution to establishing a correlation through a low-dimensional subspace), and (2) a sparse regression method (e.g., LASSO-style mapping of model onto data), both of which seem simpler and more direct than SCA.
> >
> > Scientific interpretation: The authors criticize previous methods for their insensitivity to tuning properties of individual neurons (last par of intro), and describe wiring constraints as a biological motivation (sec 2.3.1).  So the reader naturally expects that these properties and others, which are not built into a data-driven method like SCA, should emerge from the analysis.  Is there anything interesting to be noted about the structure of the recovered components?  Do they exhibit any spatial localization?  Feature localization?  Correlation to specific object categories?

---

> > > ### Author Response · Authors · 2024-12-02
> > > **Response 2 to Reviewer EmTJ (1/1)**
> > >
> > > Thank you for your continued response! You bring up excellent points, which we address here:
> > >
> > > Method:
> > >
> > > Section 2.2 outlines our motivation in using linear encoding and RSA, primarily as a way to survey different families of commonly-used alignment metrics. Linear encoding models in particular are a dominant method in measuring alignment, used in large-scale community platforms like BrainScore (Schrimpf et al 2018, bioRxiv), and one of the central concerns of this study was to address the similar alignment with DNNs and the dorsal, lateral, and ventral streams suggested by linear encoding.
> > >
> > > One advantage of using SCA and NMF-derived components is the interpretability of component response profiles and intuitive spatial layouts using anatomical projections (more details below). Nevertheless, a comprehensive comparison of SCA to other metrics would be helpful and interesting for future work. We did not include CCA given its similarity to the methods we tested here (PCA, linear regression). Finally, we note that recent studies using L1/Lasso regression dovetail nicely with the advantages we find here when using a sparse, non-negative constraint (Prince et al 2024, ICLR Workshop on Representation Alignment). Comparing SCA to CCA and/or the method used by Prince et al is an excellent suggestion for next steps.
> > >
> > > Scientific interpretation:
> > >
> > > Indeed one of the advantages of SCA and Bayesian NMF is the recovery of data-driven components without prior spatial or categorical hypotheses. Section 3.2 and Figure 4 details our results of this analysis in the dorsal, ventral, and lateral stream.
> > >
> > > Specific object categories: We find components consistently selective for a wide range of visual categories: scenes, faces, food, text, and bodies in the ventral stream; groups, motions, hands, scenes, and reach spaces in the lateral stream; and scenes and motion in the dorsal stream. We offer interpretation of component selectivity but stress that these are nonetheless reliant on the content of images in the NSD and the subjective judgements of the reader.  Similar components also emerge in neural networks trained for object recognition, which suggests some amount of feature-based clustering in component responses.
> > >
> > > Spatial localization: While we did not include an analysis of spatial layout here, prior work has used the voxel-by-component weight matrix to project and localize components in the ventral visual stream (Khosla et al 2022, Current Biology). The anatomical layout of these components align well with established functional regions of interest (fROIs) including the fusiform face area (FFA) and parahippocampal place area (PPA; Kanwisher 2010, PNAS). The voxels comprising each component cluster closely together, as might be expected from the above mentioned biological wiring constraints. These results motivated our use of NMF-derived components in SCA and are particularly striking given the lack of prior spatial or categorical information built into the method.

---

### Official Review · Reviewer_23vv · 2024-11-04

**Soundness:** 4
**Presentation:** 4
**Contribution:** 4
**Rating:** 8
**Confidence:** 4

**Summary:**

In this work, the authors apply a novel method (Sparse Component Alignment) to the problem of differentiating neural response profiles across three distinct streams of the human ventral visual pathway -- all of which are strongly theorized (after decades of intensive neuropsychological and neuroimaging work) to be functionally and representationally distinct, but which many statistical methods do not so saliently differentiate. The authors also assess the advantage of this method in the domain of brain-to-model mappings -- another area where what seems a priori to be salient functional and representational differences often fail to manifest with the statistical methods most popular for assessing them.

**Strengths:**

This paper constitutes an important contribution and meaningful step towards resolving a deep and lingering concern in computational neuroscience: How do we reconcile data-driven methods with data-derived theory? Methods developed by machine learning researchers and proponents of "hypothesis-free" (a somewhat unfortunate term) structural decomposition / discovery can sometimes clash or produce results counterintuitive to or directly at odds with observations made by experimental neuroscientists and clinicians in well-documented and amply-replicated experimental scenarios. And the issue of "resolution" tends to be one of the defining attributes of this debate. The finding by the authors that there are as-of-yet under-explored statistical methods that advance the mission to use "less biased" / less error-prone "hypothesis-free" data-driven techniques (with scare quotes galore) in a way that nevertheless accords with theoretical priors seems (in my mind) to open the door to new discovery, and far less "two steps forward, one step back"-kind-of debate along the way.

**Weaknesses:**

While I do not necessarily fault the authors for this (especially given the importance of the first-order business that involves showing this method to be intuitive on the task of differentiating response profiles across the well-separated "streams" of visual cortex), the use of only a few neural network models does not give particularly strong confidence that this method is relevant yet to the differentiation of otherwise very similar-looking brain models (another problem canonical decomposition have heretofore struggled with.)

**Questions:**

- There are a number of software packages that allow for the comparison of many different neural network models. Why not try this method on more of them? A graph that shows the differentiability of models more canonical methods versus SCA would (in my mind) be a powerful amplifier to this work.)
- The authors mention that the still-remaining non-differentiability of the lateral and dorsal streams is something that we likely need further / better / different brain data assays to meaningfully address. I am entirely on board with this argument, but one natural question that arises here is whether SCA (or any method) can be used not only for differentiation on current data, but for proposing different kinds stimuli that could maximize differentiation in future data? (Think controversial stimuli / selection).
- A bit of a follow-up on the last point (perhaps something worth addressing depending on the reaction of other reviewers). If at end of day, certain kinds of differentiation can only be achieved with the right kinds of data... doesn't this somewhat undermine one of the central motivations for using these "hypothesis-free" methods in the first place?

---

> ### Author Response · Authors · 2024-12-02
> **Response to Reviewer 23vv (1/1)**
>
> Thank you for your thoughtful feedback and thorough review of our paper. Below we respond to the questions and points of weakness you raised above:
>
> Breadth of model evaluation:
>
> You are right in critiquing our analysis of only a few DNNs. While our focus was on the alignment of standard vision models to the ventral stream, we did try to survey a range of models that varied in architecture (AlexNet vs ResNet vs ViT) and supervision (ResNet vs MoCo; ViT vs DINO).
>
> You provide excellent suggestions for future work, which should certainly apply SCA to a wider range of model architectures, training diets, and objectives to better assess its efficacy. Models trained on the kinetics dataset might show higher alignment to the lateral and dorsal stream, though motion detection (and eg. social cognition)  is far from object recognition in terms of state of the art. See also our reply to Reviewer 4
>
> Stimuli generation using SCA:
>
> Using NMF-derived components and SCA to maximally differentiate neural representations in the visual pathways and in neural networks is an interesting next step. The component response profiles in Figure 4 provide some insight into what such an approach might entail. The top images for interpretable components suggest differential categories that drive the dorsal, lateral, and ventral stream, and synthesizing stimuli may perhaps further clarify and characterize these differences.
>
> On the biases of the Natural Scenes Dataset:
>
> The choice of dataset is yet another inductive bias when training DNNs and certainly influences functional representations (see Grossman et al (2019) Nature Communications., Prince et al (2024), Science Advances).  We openly acknowledge this source of implicit bias and its effect on our claim of a ‘hypothesis-free’ method (lines 374-375). In using a massive, publicly available dataset (NSD), we hope to mitigate these concerns to some extent.

---

### Meta-Review · Area_Chair_6u51 · 2024-12-20

**Metareview:**

This paper studies human visual representation decomposition (across different visual pathways from fMRI data) and their alignment with artificial neural network counterparts. The authors use Bayesian non-negative matrix factorization (NMF) to decompose visual representations between the ventral, dorsal, and lateral streams, and introduce a new method, Sparse Component Alignment (SCA), to measure alignment between these representations and representations from (artificial) neural networks (CNNs and Vision Transformers). The authors find that ventral visual representations align significantly better with artificial neural network representations than other streams.

All reviewers positively highlighted the quality of the paper, novelty of the approach and the relevance of the results. Remaining weaknesses were well addressed during the rebuttal period, making this a solid contribution to the conference.

**Additional Comments On Reviewer Discussion:**

Reviewer consensus did not change during discussion.

---

### Decision · Program_Chairs · 2025-01-22

Accept (Spotlight)